



# Technical Note: Calibration-Free pH Sensing of Ocean and Estuarine Waters

Monica Miranda Mugica[1], Christina Day[1], Brandon McHale[1], Kay L. McGuinness[1], Gareth Lee[2], Daisy Pickup[2], Nathan S. Lawrence[1]

[1] ANB Sensors Ltd., 4 Penn Farm, Haslingfield Cambridge, CB23 1JZ, UK
[2] School of Environmental Sciences, University of East Anglia, Norwich Research Park, Norwich, NR4 7TJ, UK

*Correspondence to*: Nathan S. Lawrence (nlawrence@anbsensors.com)

**Abstract.** An electrochemical based, all solid-state, calibration-free pH sensor is presented. The sensor is targeted for monitoring the pH of ocean and estuarine environments covering a salinity range from 10-35 ppt without the need for additional salinity measurements. The sensor performance is demonstrated in both laboratory and field conditions. The field tests were conducted in an estuarine environment close to Oban in Scotland where the sensor was deployed for a period of three days. The sensor was validated against a sampled solution and tested alongside a glass pH sensor. The data highlighted the efficiency of the sensor to monitor the tidal variations of pH in the estuarine environment.

## 1 Introduction

The measurement of pH in the world's oceans is extremely important and is indispensable to researchers, industrialists, legislators and government organisations to provide an understanding of the ocean's health. The health of the ocean is impacted by various events such as pollution through industrial outfalls and environmental disasters, its absorption of $CO_2$ from the atmosphere, or through naturally occurring events such as the release of gases through subsea vents.

The pollution of rivers and oceans with chemical contaminants is one of the most critical environmental problems. Marine outfalls are commonly used for the discarding of industrial wastewater (or effluent). Effluent is often released after only partial treatment or even completely untreated and allowing the capacity of the sea for further clean up. However, this has the propensity to cause undesirable effects in the ecosystem leading to critical environmental damage (Sonune and Ghate, 2004). Effluent discharges are a major source of Contaminants of Emerging Concern (CECs) in aquatic life. Some CECs may accumulate in fish tissue (Bay et al., 2011) causing unfavourable biological effects (Tilton et al., 2002). The correct outfall system can reduce 95% of the contaminants by physical dilution, which considerably lowers the health risks. Experience has shown that a functional outfall can be a cost-effective and reliable process to release treated effluents properly with minimal environmental impact (Law and Tank, 2016).





pH is a major factor in the effluent treatment process and its monitoring throughout the process from start to outfall is extremely important in ensuring the effectiveness and performance of the process, regardless of the nature of the treatment - physical, chemical or biological (Dai et al., 2015b; pH, 2017).

Ocean acidification is a significant and unfavourable consequence of the excess carbon dioxide ($CO_2$) in the atmosphere. Prior
to the industrial revolution, carbon dioxide concentrations varied between 180 and 300 ppmv. However, today's atmospheric $CO_2$ concentration is 380 ppmv and increasing ~0.5 % per year. In the past 200 years, the world's oceans have become almost 30 % more acidic (Siegenthaler et al, 2017; Sabine et al, 2004). Water reacts with $CO_2$ to form carbonic acid ($H_2CO_3$), releasing hydrogen ions ($H^+$), and carbonate ions (Bennett, 2018; Barker and Ridgwell, 2012), causing the solution pH to decrease. If the world's oceans carry on absorbing $CO_2$ at their current rates, the pH is expected to drop to 7.8 or 7.7 by the end of this
century. A secondary issue caused by ocean acidification is the lower abundance of carbonate ions, which is key to organisms like corals or mussels to build shell and skeletons (NOAA, 2020; Fabry, 2018). Acidification decreases the ability for water to absorb additional atmospheric $CO_2$, which, if concentrations continue to increase, means that the pace of global warming will rapidly increase.

Pollution of the ocean is not only occurring through manmade events, but also through natural events such as subsea geologic formations where $CO_2$ is naturally escaping through the seafloor (e.g., hydrothermal vents). These can be difficult to detect and vary with factors such as bathymetry, hydrography, and magnitude and type of the leakage. pH is an excellent proxy for monitoring $CO_2$ levels around these vents (Botnen et al, 2015; Downing, 2014).

Today ocean pH sensors are focused on potentiometry or spectrophotometry. Potentiometry encompasses two key technologies: glass pH probes and ion sensitive field effect transistors (ISFETs). The glass pH probe is one of the most common sensors however, although special ocean glass probes are available (SeaBed, 2021; Sea & Sun, 2021), they are problematic when deployed in the ocean. They suffer from reference electrode drift, caused by changes in the environment of the reference chamber, so therefore require frequent recalibrations. They have to be kept under special storage conditions to try and keep
the reference chamber uncontaminated, and, as they are made from glass, they often suffer fragility issues. ISFET technologies are much more robust than their glass analogues and sensors have been developed specifically for the ocean environment (Takeshita et al., 2014; Kremesti, 2021). However, they still suffer from reference drift and they often have to be deployed with salinity sensors to understand reference potentials. They also face other challenges like light sensitivity and drift when multiple sensor types are co-located (Bresnahan Jr et al., 2014; Jimenez-Jorquera and Baldi, 2010; Martz et al., 2010).
Spectrophotometric based systems don't suffer from reference drift and provide very accurate pH readings (Rerolle et al., 2013). However, deployment issues around maintenance remain because of their need to mix the pH indicator dye with a seawater sample. This means they require deployment with optical dye bags which need to be replaced periodically and flowlines which can be blocked by biofouling species (Newhall et al., 2007; Rerolle et al, 2016; Chen et al., 2010).




In this paper, a new technology that exhibits several advantages from the technologies described above is presented. This relies on a redox active/pH active molecule that is combined with a redox active/pH inactive molecule to determine the pH of the seawater without the need of a calibration (Lu and Compton, 2014a). The main advantage of this novel technology is that any reference drift is tracked and accounted for in-situ, so it does not require any calibration. Moreover, the sensor does not require any special storage conditions and can be left dry, allowing for pH to be monitored in places where this has never been done

before, such as flood plain and tidal monitoring.

## 2 Technology

To overcome the drawbacks of today's pH sensing devices an innovative approach is presented encompassing a solid-state, calibration-free sensor. The sensor has been developed to be plug and play to the end user, with little maintenance and no need for regular factory recalibrations. It has a range of 2-10 pH units and an accuracy of +/- 0.05 pH units. Figure 1 shows the

image of the sensor with the sensing element, or transducer, at the front end. This transducer is a replaceable part which can be easily replaced when required (vide infra), whilst the back end utilises a submersible connector through which power and comms are passed to the control unit. On board the sensor is the electronic circuitry required to control the sensor and convert the raw measurement signal to the solution pH in a timestamped manner to the end user.

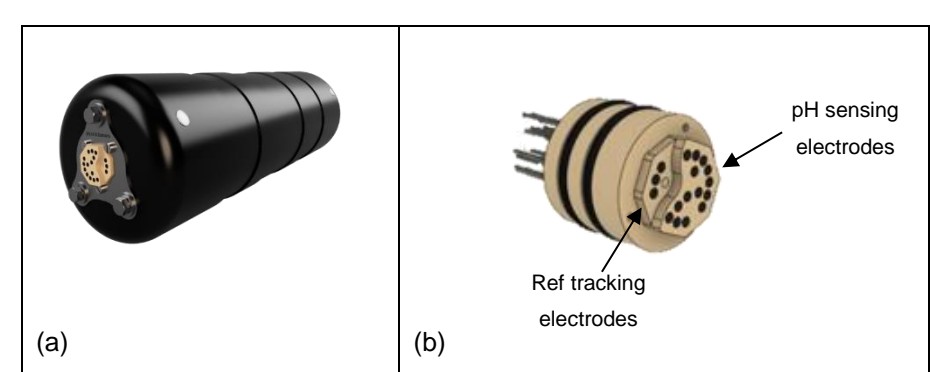


**Figure 1:** (a) The mechanical housing of the sensor including the transducer. (b) An image of the transducer depicting the recessed reference tracking electrodes and the pH sensing electrodes on the front face.

## 2.1 Transducer

This innovative pH sensor is based on a voltammetric electrochemical technique which utilizes a pH active molecule combined with a pH inactive molecule within a solid-state matrix (Batchelor-McAuley et al., 2011; Lu and Compton, 2014b). The




combination of both can determine the pH of the seawater with no need for calibration, by tracking the performance of the reference electrode through an additional in-situ electrochemical measurement (Dai et al., 2015a). The peak potential of the pH sensing electrodes moves with pH variation, allowing the detection of the change in pH. At the same time, the pH inactive electrode tracks the reference drift, permitting an in-situ re-calibration of the electrodes, and therefore, generating a calibration free pH measurement. Figure 2 shows how the peak potential moves with pH for the pH sensing electrodes (Figure 2A) whilst the reference tracker stays stable (Figure 2B), only changing in peak position when the reference is drifting.

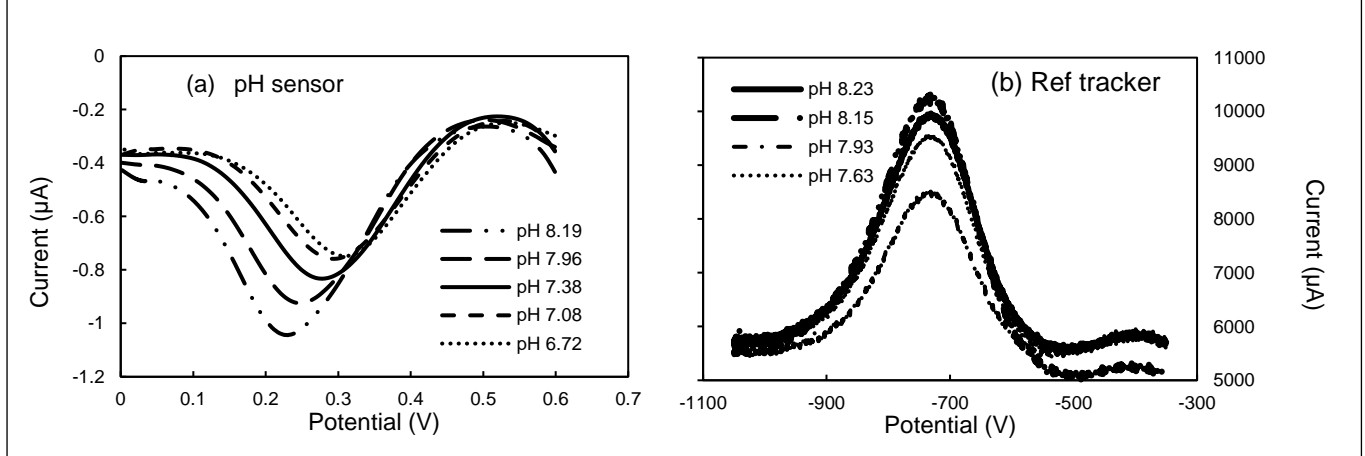

**Figure 2:** The voltammetric response collected by the sensor for both (a) the pH sensing and (b) the reference tracker electrodes.

The efficacy of this approach is shown in Figure 3. Figure 3A details the variation in peak potential of both the pH sensing and reference tracking systems as a function of time, whilst Figure 3B shows the end user result of pH as a function of time. The data highlights the robustness of the sensing system where after 10 h of deployment there is a large change in the peak potential response of both the reference tracker and pH sensor, whilst no change is seen in the pH measurement. Without the reference tracker this phenomenon would be wrongly attributed to a pH change by the end user.





**Figure 3**: (a) The plot of peak potential for the reference tracking and pH sensing elements as a function of time. (b) The final calibration free pH sensor output as a function of time.

Figure 4 shows the pH values obtained using two sensors tested in synthetic seawater (H2Ocean Natural Reef Salt, Maidenhead Aquatics, UK), where using the recipe outlined provides a pH of 8.30 at 298K. Each point represents the average across twelve pH measurements taken every five minutes of testing, along with the standard deviation of them. This data shows the reproducibility of the two sensors taken in the same tank without any calibration prior to deployment. Across the entire set of measurement an average pH value of 8.33 +/- 0.015 and 8.33 +/- 0.012, for sensor 1 and sensor 2, respectively. The data obtained is in a perfect agreement with the pH measured by a glass pH sensor (Idronaut, Italy), confirming the accuracy of the sensor and the reproducibility across different systems. The data was produced in a temperature-controlled tank (see section 3.1), at 288K.





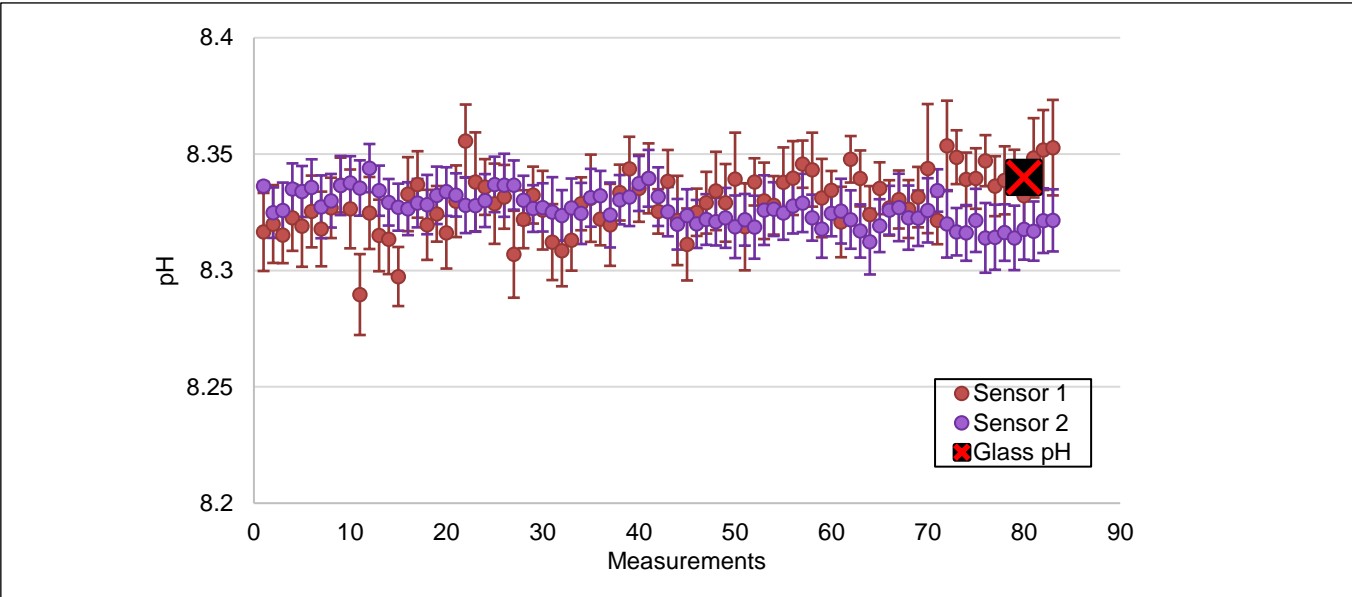

**Figure 4:** Representation of pH values obtained by two sensors in synthetic seawater at a constant temperature, compared with the pH measured by a glass pH probe. Each measurement corresponds to twelve real pH measurement over 5 minutes testing.

## 2.2 Lifetime and Maintenance

The lifetime of the sensor is dependent on the number of measurements that each electrode records and not on the deployment time of the sensor. The repetitive voltammetric measurements on each electrode ages the electrode until the peak potential can no longer be deciphered and maintenance is required. It is for this reason that there are a number of electrodes on the front surface of the sensor, (twelve electrodes are pH sensing, three are reference tracking and these are combined with two counter electrodes and a silver rod which acts as the conventional electrochemical reference electrode) in order to maximise the lifetime of the sensor before maintenance is required.

The sensor has an onboard processor which assesses the voltammetric profile of each electrode. The response of each electrode is analysed after each measurement to ensure an accurate response is observed, when this is not the case the health number is increased, to alert the end user that maintenance is required.



Maintenance of the sensor is through an abrasion of the transducer surface using an abrasion block (wet and dry paper, 320 grade). After the abrasion the sensor can be redeployed, and the health of the sensor returns to its original value. The transducer displays up to 15,000 measurements between abrasions and will last for approximately 25-30 abrasions before it will need replacing. The sensor has been designed so that the replacement of the transducer is very simple for the end user.

## 2.3 Electronics and Communication

The sensor operates in either autonomous (stand-alone) or user-controlled mode through an RS232/485 communication protocol. In autonomous mode the sensor is connected to a power source and operates at a pre-set measurement rate. The maximum frequency for displaying a pH reading is approximately 15 sec, with the possibility of decreasing the measurement rate depending on the end user requirements. The data is saved to an SD card and can be downloaded after deployment through a data retrieval unit via a USB connection to a PC.

In user-controlled mode the user defines the measurement rate and initiates the scan and shutdown commands. The data is returned to the control unit in real time. Table 1 summaries the supply voltage and current draw when the sensor is in various modes.

**Table 1:** A summary of the supply voltage and power consumption of the sensor.

| | | |
|---|---|---|
| **Supply Voltage** | **External** | **6.5 to 20 VDC** |
| Power Consumption | Sampling | ca. 110 mA |
| | Sleep | 5 mA |
| On board Storage | | 8 GB |

The pH output is calculated using an inbuilt algorithm which combines a knowledge of the peak potential of both the pH sensing and reference tracking elements and the temperature to produce the pH. The temperature is measured using the onboard k type thermocouple. An example of the sensor output is shown in 5. After the system details (firmware version, serial number, date, time, scanning mode) the sensor outputs the date/time, pH, electrode number, temperature and health. As mentioned above, the health of the electrode corresponds to its condition and varies from 0 to 9. When the health of the majority of electrodes reaches 9, the transducer will require an abrasion.



175

```
****************************************************
                   ANB SENSORS
     Interface Firmware Version : V11.3
     Driver Firmware Version    : V7.0
     System Serial Number       : 000035
     System Date (MM/DD/YY)      : 12/30/21
     System Time (24 Hour)       : 10:05
     System Style                : Controlled Scanning
     Scanning Interval           : 15 Minutes
****************************************************
scan
$ANB,05B8,0,000035,2021:12:30:10:06:02
$ANB,C40A,0,2021:12:30:10:07:26,$$.$$$,12,281.300,3
$ANB,E0E8,0,2021:12:30:10:07:49,07.505,04,281.300,0
$ANB,9E0D,0,2021:12:30:10:08:12,07.476,08,281.300,0
$ANB,1AD4,0,2021:12:30:10:08:35,07.442,10,281.300,0
$ANB,707F,0,2021:12:30:10:08:57,07.409,02,281.300,3
$ANB,1803,0,2021:12:30:10:09:20,07.437,05,281.400,0
$ANB,E74D,0,2021:12:30:10:09:43,07.430,07,281.400,0
$ANB,9754,0,2021:12:30:10:10:06,07.470,09,281.400,0
$ANB,686C,0,2021:12:30:10:10:29,07.524,03,281.400,0
$ANB,00F2,0,2021:12:30:10:10:53,07.499,06,281.400,0
$ANB,2197,0,2021:12:30:10:11:16,07.481,11,281.400,0
```

**Time:** year:month:day:hour:minute:sec

(e.g.: 2021:12:30:10:07:49)

**pH:** X.XXX

(e.g.: 7.505)

**Electrode:** electrode from which pH is cycled

(e.g.: 04)

**Temperature:** XXX.XXX K

(e.g.: 281.300 K)

**Health:** Scaled from 0-9

(e.g.: 0)

**Figure 5:** Sensor output from the terminal program.

180

## 3. Experimental

### 3.1 Laboratory Tests

Laboratory tests were conducted in temperature controlled stirred solutions. The solution was produced using synthetic sea salts (H2Ocean Natural Reef Salt, purchased from Maidenhead Aquatics, UK) diluted to the correct concentration using deionised water. $CO_2$ (BOC, 99%) was bubbled into the solution periodically to manipulate the pH. Temperature control was obtained through a temperature exchange loop placed inside a temperature-controlled chamber. The exchange loop temperature was manipulated through an external temperature-controlled bath (Fisher Scientific Isotemp 4100 R20 Refrigerated/Heated Bath Circulator) and held at 15°C for the time of the experiment. The solution was stirred using a Teflon overhead stirrer and the temperature measured using a k type thermocouple sheathed in polymer housing. The sensors were immersed in the tank and connected to a control unit via a SubConn cable. The sensors were powered using a 12 V battery supply and each sensor used was switched on and off during the course of the experiment to ensure there was no bias in the experiment. The solution was sampled periodically throughout the course of the experiment and the pH measured using a calibrated glass ocean pH probe (Idronaut, Italy).



## 3.2 Field Trials

195    The sensor was tested in Ardmucknish bay, located in Dunbeg, a village located outside Oban, Scotland (6). Dunbeg is home
to the one of the leading marine science centres in the UK, the Scottish Association for Marine Science (SAMS). Ardmucknish
bay is a wide-open, south facing bay, opposite to the island of Mull.

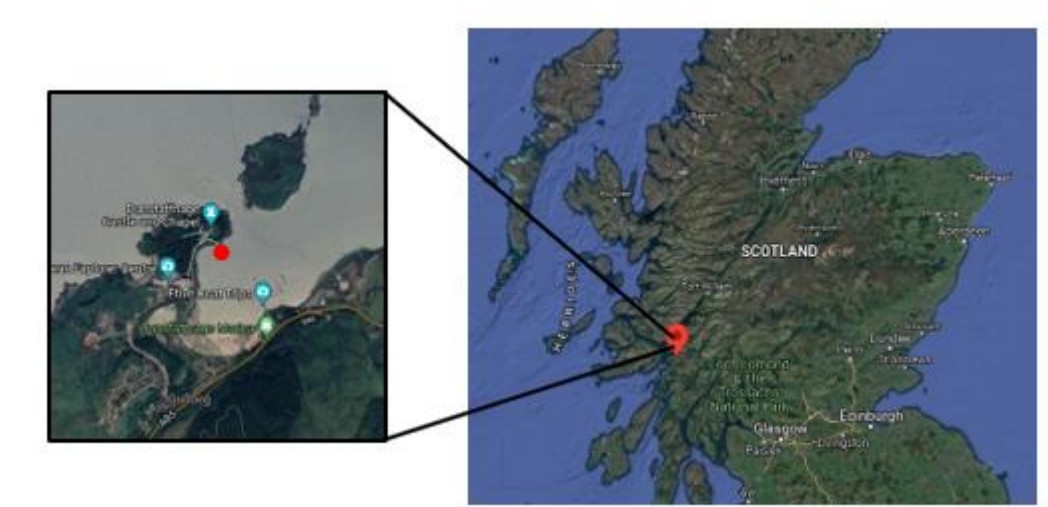

200    **Figure 6:** Sensor testing location: Ardmucknish bay, Dunbeg, Scotland.

© Google Maps

## 4 Demonstration of deployments

### 4.1 Laboratory Testing

205    Testing of the sensor in the laboratory has focused on lifetime stability and the ability of the sensor to respond in real time to
changes of pH in seawater solutions. Three different sensors were used to monitor the pH of a seawater solution for a week,
and during this time $CO_2$ was added to the solution sporadically to vary the pH. 7 details the plot of pH as a function of time
for all three sensors. The open circles represent the times at which samples of solution were taken and the pH measured with
the freshly calibrated glass pH probe. The data shows excellent agreement both between the three sensors and between the
210    sensors response and the pH measured with the glass pH probe. The sensors result lay within the specifications outlined with
respect to intra sensor repeatability and validation against the externally calibrated probe.

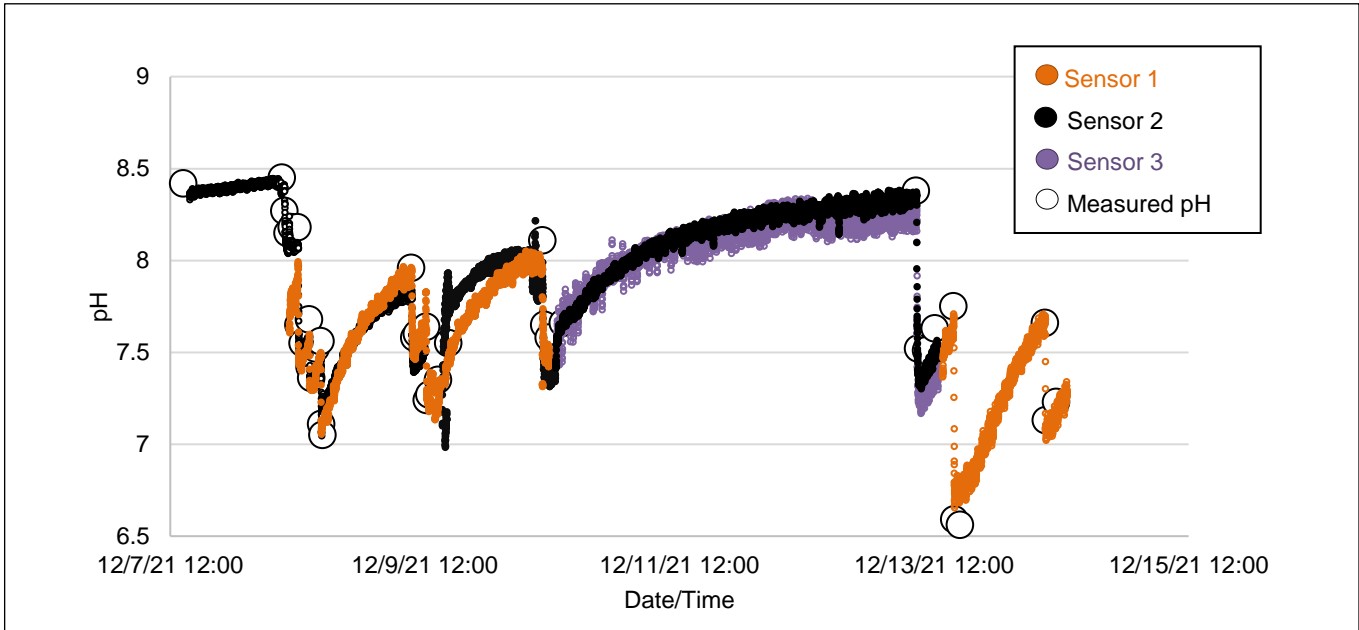

215

**Figure 7:** A plot of pH as a function of time for three sensors placed in a temperature controlled synthetic seawater solution to periodic injections of $CO_2$, along with samples measured with a calibrated glass pH electrode (open circles).

### 4.2 Field Trials

The Field Trials were conducted over a period of 3 days (8th - 10th September 2021), with a pre-set delay time of 15 min. between measurements. In this case the sensor was placed in autonomous mode, powered directly from a 12 V power supply with no interface control. The data was saved to the internal memory of the sensor and downloaded after deployment. 8 details the data obtained from the sensor through a plot of pH variation (purple) and temperature (black) as a function of time over the 3 days. During the course of the trials a water sample (ca. 250 mL) was taken in a plastic bottle and a pH of 7.73 was measured using a calibrated glass pH electrode. The data output from the sensor after 36 hr. of measuring on a 15 min. measurement interval showed the bi-daily variations in pH ranging from 7.7 up to 7.9, consistent with the natural ebb and flow of the tidal waterways at Oban. At high tide, the river water mixes with the incoming seawater, producing a higher pH value. The pH variation with tides was quite noticeable, a result of the high tidal coefficient in those 3 days. Tidal coefficients are referred to as the difference in water height between high and low tides, on a scale of 20 to 120. The sensor was deployed from the 8th - 10th September when high tidal coefficients - 101, 99 and 91 occurred (Oban, 2021), explaining why such significant variations in pH were observed.




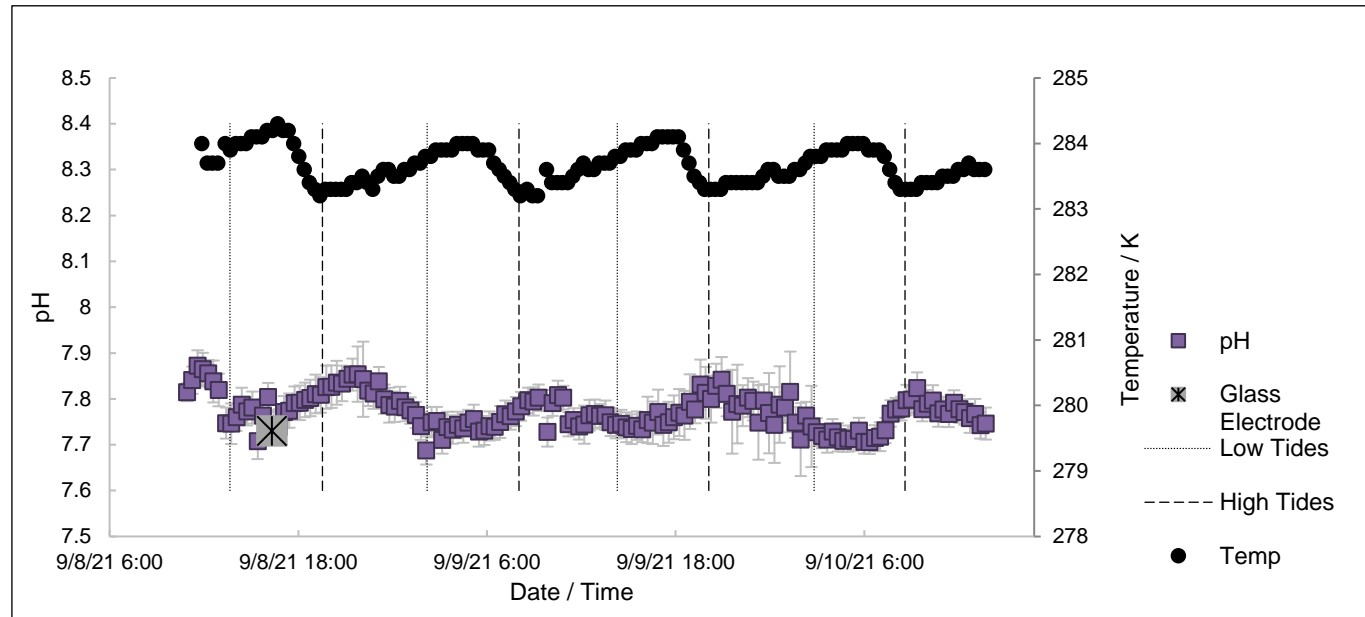

**Figure 8:** pH and temperature measurement obtained by the sensor, correlated with high and low tides in Ardmucknish bay, along with the pH of a water sample measured by a calibrated glass electrode (*).

During the final day of the trial a glass electrode was deployed alongside the solid-state sensor. The corresponding data comparing the solid-state sensor response to the glass electrode is shown in Figure 9. It can be clearly seen the glass electrode displayed a very unstable pH reading, with values ranging between 7.50 and 7.90 in pH, endorsing the advantage of using solid-state against the glass electrode for deployment in sea water.

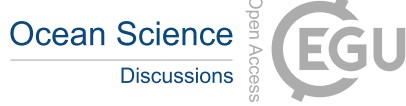

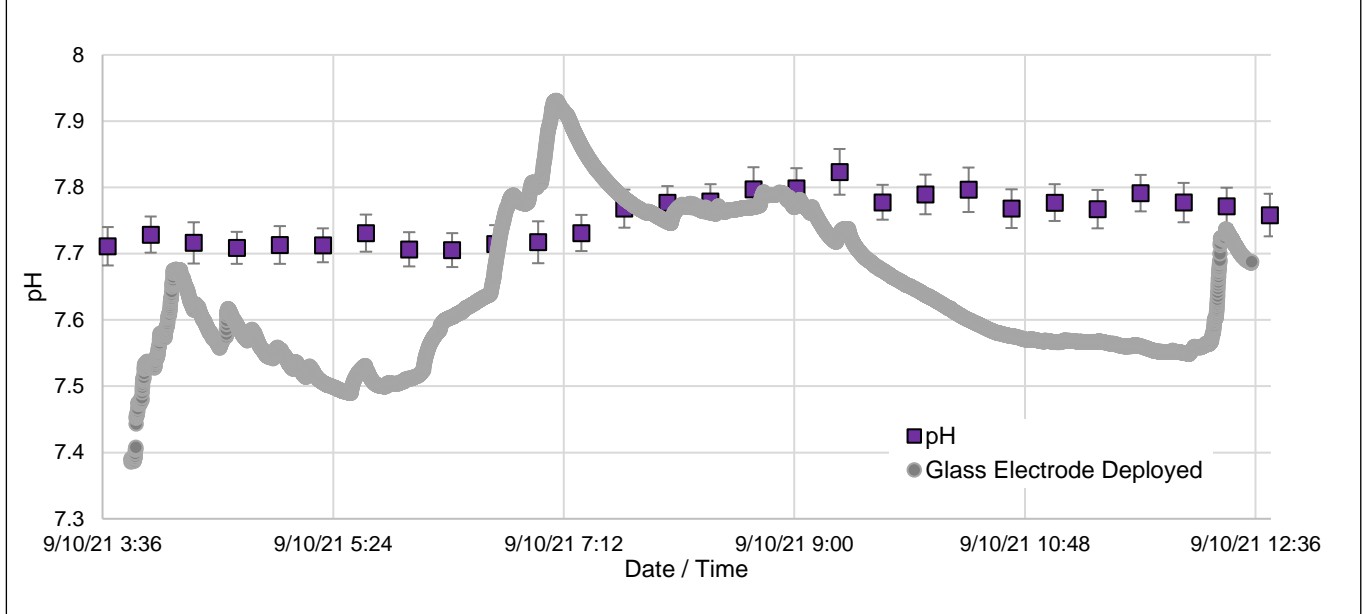

**Figure 9:** A plot of pH as a function of time for the sensors deployed in Ardmucknish Bay for both the solid-state pH sensor and an oceanographic glass-based pH sensor.


### 5. Conclusions

The results demonstrate the use of a new solid-state, calibration-free pH sensor in the monitoring of estuarine and ocean waters. The sensor was shown to respond effectively to pH in laboratory environments whereby the pH of the sea water solution was manipulated by the sporadic injection of $CO_2$ into the mixed solution. Sensors were run in conjunction to each other showing

the intra reproducibility of the sensors and compared to a sample of solution whose pH was measured using a calibrated glass electrode. Excellent agreement was observed throughout the five-day experiment. The field trials were conducted in an estuarine environment close to Oban in Scotland where the sensor was deployed for a period of three days. The sensor was validated against a sampled solution and tested alongside a glass pH sensor system. The data highlighted the efficiency of the sensor to monitor the tidal variations of pH in the estuarine environment.

**Acknowledgements**

This project has received funding from the European Union's Horizon 2020 research and innovation program under grant agreement No 82297. ANB Sensors gratefully acknowledges funding this work through Innovate UK grant funding, Grant no. 133171. Innovate UK is the UK's innovation agency. It works with people, companies and partner organisations to find and



drive the science and technology innovations that will grow the UK-economy. The authors would like to thank the team at
SAMS for allowing them to conduct the tests at their facilities.

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



Underwater measurement solutions: Sea & Sun Technology - Multiparameter Probes (sea-sun-tech.com). Last access: 20 October 2021.

Why pH is important?: Why pH is important? (aperainst.com), 15 November 2017. Last access: 17 October 2021.