# Peer review of "Technical Note: Calibration-Free pH Sensing of Ocean and Estuarine Waters"

_Ocean Science, 2021_

## Author Comment (AC1)

This manuscript is badly organized with a lot of important information left out. There are a lot of issues with the experiment, and the results do not support the author's points at all. This manuscript should be rejected. The authors need to re-design and re-do all the experiments, and follow the instructions of Ocean Science journal to re-write the manuscript in the correction structure (Introduction, Experimental, Results and Discussion, Conclusion). The manuscript has been restructured in line with the journal instructions and consistent with previously published manuscripts within the journal.

- 1. The background info about the importance of pH in the ocean (line 1 line 48) seems to be a bit tedious, which needs to be consolidated. [1] The information has been consolidated.
- 2. In the introduction section, it'd be very useful to list the accuracy requirements of each application (e.g. ocean acidification research, routine water monitoring, aquarium management, and aquaculture). [2] Information about accuracy requirements has been added.
- 3. The author should mentioned that the state-of-the-art lab method for seawater pH measurement is the indicator dye-based spectrophotometric method. A table of the accuracy and precision of each method (i.e. lab based spectrophotometric method, spectrophotometry-based sensors, potentiometry-based sensors, and the solid state sensor) would be very useful here. Data pertaining to the accuracy and precision of each technique is readily available and so we feel a detailed comparison is not necessary in the current manuscript.
- 4. The author should write about the chemistry behind the solid state pH sensor either in the 'Introduction' section, which is important for a technic note like this. [3] The chemistry behind the sensor has been briefly mentioned. The scientific papers about both pH sensing and reference tracking chemistry previously published have been cited.
- 5. Some contents of the 'Technology' section belongs to the 'Introduction', and the rest should be moved to the 'Result and Discussion'. The manuscript has been restructured.
- 6. Line 91 Line 92: Do you mean the 'peak height' changes for reference tracker electrode? No, it is related to the actual peak potential of the reference electrode.
- **7.** The titles of Figure 2 need to be consistent with the figure caption ( '(a) pH sensing electrode', '(b) reference tracking electrode'). [4] Adapted.
- 8. Figure 3(a): Need to add parameter name and unit for Y-axis. Name and unit have been added.

- 9. How is experiment done for Figure 3? What pH buffer solution was used? Was it a fresh water buffer or seawater buffer? How was the temperature controlled? What was the temperature? [5] The sensor was just deployed in synthetic seawater, and the resulting pH was calculated with the combination of both pH sensing and reference tracker electrodes. Temperature controlled information has been added.
- 10. Need to use a smaller Y-range in Figure 3b. I don't know how the experiments was done, but such pH variation is actually too big for most ocean acidification research. Smaller Y-range has been used for Figure 3B (newly Fig. 4B). The pH variation shows an accuracy of +/-0.05 in pH, as presented in the paper. However, this experiment was mainly performed to show the behaviour of the reference tracking electrode when the Ag/AgCl reference drifted, proving the ability of our system to self-calibrate. More controlled experiments were performed (Fig 4 newly Fig 5) to study the accuracy of the system and the ability to use it as an ocean sensor.
- 11. Line 120: You need to use pH buffer solution for such experiment. Such as tris-buffer in synthetic seawater. How is the temperature controlled? [5] We do not use pH buffers solutions for calculating the pH as the drift is corrected by the reference tracking electrode. Thus, the system does not need to be calibrated.
- 12. Line 124: It is wrong to use a glass electrode as the gold-standard for such experiment, since the glass electrode itself could be drifting. You need to use the state-of-the-art lab-based spectrophotometric method for the validation. And the validation measurements need to be taken throughout the whole period. A full-ocean glass pH sensor was used as a standard because it is specifically for ocean monitoring applications, and no drift was assumed as it was freshly calibrated. The reason of not using it through the whole experiment is the actual drift; after leaving the glass pH electrode for that long it would very probably drift and not give any realistic value, at the contrary of being freshly calibrated.
- 13. Figure 4: Use time rather than measurements as the X-axis
- 14. If the temperature and the pH of the testing solution were stable, the result in Figure 4 shows: (1) there are really big and non-consistent differences between the two solid state sensors; and (2) for an individual sensor the readings were not stable at all. And the conclusion will be this solid-state sensor is really bad and not suitable for ocean acidification research. Both systems show pH readings between 8.30 and 8.35, with a standard deviation of +/- 0.045 and +/-0.031 per sensor, and +/- 0.015 and +/- 0.012 between the 12 pH readings per measurement per sensor. Based on this numbers, the sensor would be suitable for ocean research as the accuracy required is +/-0.05 pH units.

- 15. The 'Experimental' section should be right behind the 'Introduction' Section. Structure of the paper has been adapted, and the 'Experimental' section is now after the 'Introduction'.
- 16. Some contents of the 'Demonstration of deployments' should be in the 'Experimental' section, and the rest should be part of 'Result and Discussion' section. Structure of the paper has been adapted. 'Results and Discussion' has been named instead of 'Demonstration of deployments'.
- 17. Section 3.1 and 4.1: Again, it is wrong to use a glass electrode as the goldstandard for such experiment, since the glass electrode itself could be drifting. You need to use the state-of-the-art lab-based spectrophotometric method for the validation. A freshly calibrated full-ocean glass pH sensor can be trusted for a pH measurement point (however, not for a deployment, as demonstrated in figure 9).
- 18. Figure 5 is unnecessary. The aim of this figure is to show the sensor's output from the program, and explain the health number of the sensor, which tells the end user when maintenance in needed.
- 19. Texts in Figure 6 are very blurry. It has been changed.
- 20. Section 4.1: What is the point to change sensors during the long-term experiment? It'd be better to use all three sensors for the entire experiment period so that both accuracy and the reproducibility of different sensors can be evaluated. The idea of changing the sensors was to compare a third one between the initial two at a different point or lifetime of the experiment. This indicates that sensor 1 maintains its calibration during the course of the experiment compared to the freshly prepared sensors.
- 21. Figure 7: It would be more useful to plot the sensor values against the pH measured by standard method, do a linear regression and give the statistics (e.g. slope, intersect, r square, and RMSE). The aim of Figure 7 is to show the response of the sensor when injecting CO2 into the system several times, while the seawater recovers the pH value in between additions. The response of the sensor against a constant variation in pH levels is shown.
- 22. Section 4.2: It is wrong to use a glass electrode as the gold-standard for such experiment, since the glass electrode itself could be drifting. You need to use the state-of-the-art lab-based spectrophotometric method for the validation. A freshly calibrated full-ocean glass pH sensor can be trusted for a point pH measurement validation.
- 23. Figure 8: only one validation point is far from enough. Validation measurements need to be taken throughout the entire period. After the validation point was taken, the glass pH sensor was deployed into the seawater, with the idea of validating all the test. However, the glass pH data was unstable (as shown in Figure 9). Access to the field location facilities meant the experiment could not be repeated.

- 24. Figure 9: This figure doesn't mean anything. First of all, there is no validation and we don't know if the solid state sensor or the glass electrode are correct. Secondly, 'more stable' doesn't mean it is correct. In contrast, it could mean it is not sensitive enough. As mentioned in comment 14, a non-stable pH reading would mean unsuitable pH probe for ocean, which is what the glass pH electrode shows. The pH value calculated with our sensor matched the validation of the calibrated glass pH electrode. Knowing our sensor is measuring the right pH value, and because of the expected response of the sensor against the tidal pH variation originated by the inflow of the fresh SW into the bay.
- 25. Section 5: The experiments don't support the author's claims at all. Based on the data presented in this manuscript, it seems that this sensor is not suitable for ocean acidification research. We believe the sensor has proved to achieve the accuracy presented (0.05 in pH) and to be sensitive enough to follow the tidal pH variations, as well as a good reproducibility between different sensors. The accuracy of the sensor has been successfully validated using a freshly calibrated full-ocean pH probe.

---

## Author Comment (AC2)

**Technical Note: Calibration-Free pH Sensing of Ocean and Estuarine Waters**

Monica Miranda Mugica[1], Christina Day[1], Brandon McHale[1], Kay L. McGuinness[1], Gareth Lee[2], Daisy Pickup[2], Nathan S. Lawrence[1]

[1] ANB Sensors Ltd., 4 Penn Farm, Haslingfield Cambridge, CB23 1JZ, UK
[2] School of Environmental Sciences, University of East Anglia, Norwich Research Park, Norwich, NR4 7TJ, UK

*Correspondence to*: Nathan S. Lawrence (nlawrence@anbsensors.com)

**Abstract.** An electrochemical based, all solid-state, calibration-free pH sensor is presented. The sensor is targeted for monitoring the pH of ocean and estuarine environments covering a salinity range from 10-35 PSU without the need for any salinity measurements. The sensor performance is demonstrated in both laboratory and field conditions, showing a pH of 8.33 in two sensors, against a pH of 8.34 of a freshly calibrated glass pH sensor. Excellent precision is also shown, exhibiting a pH of 8.33 +/- 0.015 and 8.33 +/- 0.012 across multiple measurements. The field tests were conducted in an estuarine environment close to Oban in Scotland where the sensor was deployed for a period of three days. The sensor was validated against a sampled solution that measured a pH of 7.74, which was equivalent to the pH 7.74 obtained from ANB's pH sensor at that exact time, and it was tested alongside a glass pH sensor during deployment, demonstrating its stability over 3 days of testing. The data highlighted the ability of the sensor to monitor the tidal variations of pH in the estuarine environment.

**1 Introduction**

The measurement of pH in the world's oceans is extremely important and is indispensable to researchers, industrialists, legislators and government organisations to provide an understanding of the ocean's health. The health of the ocean is impacted by various events such as pollution through industrial outfalls and environmental disasters, its absorption of $CO_2$ from the atmosphere, or through naturally occurring events such as the release of gases through subsea vents.

pH is a major factor in the effluent treatment process and its monitoring throughout the process from start to outfall is extremely important in ensuring the effectiveness and performance of the process, regardless of the nature of the treatment - physical, chemical or biological (Dai et al., 2015b; pH, 2017).

Ocean acidification is a significant and unfavourable consequence of the excess carbon dioxide ($CO_2$) in the atmosphere. Prior to the industrial revolution, carbon dioxide concentrations varied between 180 and 300 ppmv. However, today's atmospheric $CO_2$ concentration is 380 ppmv and increasing ~0.5 % per year. In the past 200 years, the world's oceans have become almost 30 % more acidic (Siegenthaler et al, 2017; Sabine et al, 2004). Water reacts with $CO_2$ to form carbonic acid ($H_2CO_3$), releasing

hydrogen ions (H⁺), (; Barker and Ridgwell, 2012), causing the solution pH to decrease. Previously it was believed that dissolved chemicals carried by the rivers to the ocean were enough to keep the ocean's pH stable and so the impact of rising $CO_2$ levels would be negated. However, the rate of dissolution of carbon dioxide into the ocean is so rapid, that the natural buffering effect has been overwhelmed, resulting in a rapid decrease of ocean pH (Bennett, 2018). [1] If the

35 world's oceans carry on absorbing $CO_2$ at their current rates, average global ocean surface pH is expected to drop to 7.8 or 7.7 by the end of this century creating the most acidic ocean for the past 20 million years (Caldeira and Wicket, 2005). [1]

A secondary issue caused by ocean acidification is the lower abundance of carbonate ions, which is key to organisms like corals or mussels to build shell and skeletons (NOAA, 2020; Fabry, 2018). Many chemical reactions are sensitive to small

40 changes in pH which can have detrimental effects on marine life, affecting reproduction and growth (Doney et al., 2009). This is a key reason why pH is one of the most important ocean water quality parameters (Hickin, 1995). [1]

Excess carbon dioxide in the atmosphere, is causing higher ambient temperatures, especially in the poles, where many glaciers are now rapidly melting. These melting glaciers impact ocean currents and rises sea levels. If $CO_2$ discharge continues to

45 increase, the current rate of melting ice is expected to double by the end of the century, and Antarctic melt alone will raise the global sea level by 5 feet (Hancock, 2022; Stone, 2021). [1]

[revised manuscript text omitted]

**3.1 Transducer**

This innovative calibration-free pH sensor is based on a voltammetric electrochemical technique, where a time-dependent
125 potential is applied to each electrode in the transducer and the resulting current is measured as a function of that potential
(Bard, 2000). The pH is calculated using the resulting peak potential, as it is linearly associated with the pH. The sensor utilizes
a pH active molecule combined with a pH inactive molecule within a solid-state matrix. 2'-hydroxyflavanone is used as the
electroactive polymer for the pH sensing part (Miranda, 2022), and anthraquinone – modified carbon electrode (Sisodia, 2022)
as the pH insensitive sensor. [3] The electrochemical response of both species can determine the pH of the seawater with no
130 need for calibration, by tracking the performance of the reference electrode (pH inactive molecule) through an additional in-
situ electrochemical measurement (Dai et al., 2015a). As mentioned, the peak potential of the pH sensing electrodes moves
with pH, allowing the determination of pH. At the same time, the pH inactive electrode tracks the reference drift, and therefore,
generates a calibration free pH measurement. Figure 3 shows how the peak potential shifts to lower potentials as the pH of the
solution increases for the pH sensing electrodes (Figure 3A) whilst the reference tracker potential stays stable (Figure 3B),
135 only changing in peak position when the reference is drifting. Figure 3 was obtained by testing both pH sensing and reference
trackers in synthetic seawater (H2Ocean Natural Reef Salt, Maidenhead Aquatics, UK). The pH of the solution was decreased
by the addition of $CO_2$, allowing the response of both the pH sensing and reference sensing elements to be monitored. The pH
was measured using a full ocean glass pH probe (Idronaut, Italy), which it was calibrated using standard buffers. When the pH
sensing electrodes and reference tracker are combined, subtracted values can be calculated, and based on the calibration
140 algorithm shown in Fig. 3C, the resulting pH can be determined. The calibration of the sensor shows a sensitivity of 54.8
mV/pH unit in synthetic seawater, overlaying the calibration plot obtained for pH 4, pH 7 and pH 9 IUPAC standard buffers,
with a sensitivity of 56.5 mV/pH unit [1] [10] (Miranda Mugica et al., 2022)

[Figure]

**Figure 3:** The voltammetric response collected by the sensor for both (a) the pH sensing (frequency=100 Hz, step potential=1 mV, amplitude=40 mV) and (b) the reference tracker electrodes (frequency=300 Hz, step potential=0.5 mV, amplitude=80 mV) in synthetic seawater/$CO_2$ system, and (c) the resulting calibration trendline in synthetic seawater and IUPAC standard buffers, using the subtracted values. pH values were measured using a calibrated glass pH probe.

The efficacy of this approach is shown in Figure 4, where the sensor was deployed in a temperature-controlled seawater tank, at 293K. [5] Figure 4A details the variation in peak potential of both the pH sensing and reference tracking systems as a function of time. After 10 h of deployment, a large change in peak potential was observed in the pH sensing system. Nevertheless, the reference tracker exhibited the same change in potential, meaning that the Ag/AgCl redox couple was impacted due to a change in the environment of the reference electrode. The tracking of the reference electrode allows the system to correct the internal drift of the reference electrode. Figure 4B shows the end user result of pH as a function of time, displaying a pH 8.07 +/- 0.039 over a 14 h experiment, highlighting the robustness of the sensing system where no change is seen in the pH measurement owing to the reference electrode tracker correction, with no need of re-calibration. Without the reference tracker this phenomenon may be wrongly attributed to a pH change by the end user.

[Figure]

**Figure 4**: (a) The plot of peak potential for the reference tracking and pH sensing elements as a function of time. (b) The final calibration free pH sensor output as a function of time.

Figure 5 shows the pH values obtained using two sensors tested simultaneously in synthetic seawater, in a temperature-controlled tank [6], (see section 2.1) at 288K, where using the recipe outlined provides a pH of 8.30 at 298K. The sensor scans the reference electrode first, to track any changes in the reference electrode potential, and then the twelve pH sensitive electrodes in the transducer head, taking ca.15 seconds for each pH measurement. Each data point represents the average of the twelve pH measurements over five minutes testing. This data shows the reproducibility of the two sensors taken in the same tank without any calibration prior to deployment. Across the entire set of measurements an average pH value of 8.33 +/- 0.015 and 8.33 +/- 0.012 for sensor 1 and sensor 2, respectively, were obtained. Without averaging the twelve pH measurements, the accuracy for each sensor was 8.33 +/- 0.04 and 8.33 +/- 0.03. The data obtained is in a excellent agreement with the pH 8.34 measured by a freshly calibrated full ocean glass pH sensor as well as with the expected theoretical pH of the synthetic seawater. Even though glass pH electrodes suffer from reference drift and are not ideal for deployment conditions

due to their fragility, a one-point measurement can be trusted when freshly calibrated using standard buffers. For that reason, the accuracy of the sensor can be confirmed, along with the reproducibility across different systems.

[Figure]

195

**Figure 5:** Representation of pH values obtained by two sensors in synthetic seawater at a constant temperature, compared with the pH measured by a glass pH probe. Each measurement corresponds to the average of twelve pH measurements over five minutes testing.

**3.2 Lifetime and Maintenance**

200

The lifetime of the sensor is dependent on the number of measurements that each electrode records and not on the deployment time of the sensor. The electrodes in the transducer are repetitively cycled and the continual voltammetric measurements on each electrode decays the response of each electrode. This is seen through a loss in the voltammetric peak size until the peak potential can no longer be deciphered. At this point the sensor notifies the end-user maintenance is required. It is for this reason

205  that there are a number of electrodes on the front surface of the transducer so as to maximise the lifetime of the sensor before maintenance is required.

The sensor has an onboard processor which applies the one-time voltammetric sweep parameters to the electrochemical reaction occurring at each electrode. The resulting current received at the electrode due to the electrochemical reaction is then recorded, the data is analysed to determine the potential at which maximum current flows, which is then used to calculate the

210 corresponding pH value. The response of each electrode is analysed after each measurement to ensure an accurate response is observed, when this is not the case the health number is increased, to alert the end user that maintenance is required.

Maintenance of the sensor is through an abrasion of the transducer surface using an abrasion block (wet and dry paper, 320 grade). After the abrasion the sensor can be redeployed, and the health of the sensor returns to its original value. The transducer
215 displays up to 15,000 measurements between abrasions and will last for approximately 25-30 abrasions before it will need replacing. The sensor has been designed so that the replacement of the transducer is very simple for the end user.

**3.3 Electronics and Communication**

The sensor operates in either autonomous (stand-alone) or user-controlled mode through an RS232/485 communication protocol. In autonomous mode the sensor is connected to a power source and operates at a pre-set measurement rate. The
220 maximum frequency for displaying a pH reading is approximately 15 sec, with the possibility of decreasing the measurement rate depending on the end user requirements. The data is saved to an SD card and can be downloaded after deployment through a data retrieval unit via a USB connection to a PC.

In user-controlled mode the user defines the measurement rate and initiates the scan and shutdown commands. The data is
225 returned to the control unit in real time. Table 1 summaries the supply voltage and current draw when the sensor is in various modes.

230

**Table 1:** A summary of the supply voltage and power consumption of the sensor.

| | | |
|---|---|---|
| **Supply Voltage** | **External** | **6.5 to 20 VDC** |
| Power Consumption | Sampling | ca. 110 mA |
| | Sleep | 5 mA |
| On board Storage | | 8 GB |

The pH output is calculated using an inbuilt algorithm based on the equation presented in Fig. 3C ($y = -0.0548x + 0.6794$
235 where $x$ corresponds to pH and $y$ to the peak potential (V)), which combines a knowledge of the peak potential of both the pH sensing and reference tracking elements. As mentioned before, a potential is applied to each electrode and the resulting current

is measured as a function of that potential, obtaining the resulting voltametric response, which will provide the peak potential value needed for the calculation of the pH. Temperature is also needed for the algorithm to calculate the pH. The temperature is measured using the onboard k type thermocouple. An example of the sensor output is shown in Fig. 6. After the system details (firmware version, serial number, date, time, scanning mode) the sensor outputs the date/time, pH, electrode number, temperature and health. As mentioned above, the health of the electrode corresponds to its condition and varies from 0 to 9. When the health of the majority of electrodes reaches 9, the transducer will require an abrasion.

```
* * *
                 ANB SENSORS
     Interface Firmware Version : V11.3
     Driver Firmware Version    : V7.0
     System Serial Number       : 000035
     System Date (MM/DD/YY)      : 12/30/21
     System Time (24 Hour)       : 10:05
     System Style               : Controlled Scanning
     Scanning Interval          : 15 Minutes
* * *
scan
$ANB,05B8,0,000035,2021:12:30:10:06:02
$ANB,C40A,0,2021:12:30:10:07:26,$$.$$$,12,281.300,3
$ANB,E0E8,0,2021:12:30:10:07:49,07.505,04,281.300,0
$ANB,9E0D,0,2021:12:30:10:08:12,07.476,08,281.300,0
$ANB,1AD4,0,2021:12:30:10:08:35,07.442,10,281.300,0
$ANB,707F,0,2021:12:30:10:08:57,07.409,02,281.300,3
$ANB,1803,0,2021:12:30:10:09:20,07.437,05,281.400,0
$ANB,E74D,0,2021:12:30:10:09:43,07.430,07,281.400,0
$ANB,9754,0,2021:12:30:10:10:06,07.470,09,281.400,0
$ANB,686C,0,2021:12:30:10:10:29,07.524,03,281.400,0
$ANB,00F2,0,2021:12:30:10:10:53,07.499,06,281.400,0
$ANB,2197,0,2021:12:30:10:11:16,07.481,11,281.400,0
```

**Time:** year:month:day:hour:minute:sec

    (e.g.: 2021:12:30:10:07:49)

**pH:** X.XXX

    (e.g.: 7.505)

**Electrode:** electrode from which pH is cycled

    (e.g.: 04)

**Temperature:** XXX.XXX K

    (e.g.: 281.300 K)

**Health:** Scaled from 0-9

    (e.g.: 0)

**Figure 6:** Sensor output from the terminal program.

**4 Results and Discussion**

**4.1 Laboratory Testing**

Testing of the sensor in the laboratory has focused on lifetime stability and the ability of the sensor to respond in real time to changes of pH in seawater solutions. Three different sensors were used to monitor the pH of a seawater solution for a week, and during this time $CO_2$ was sporadically bubbled into the solution to vary the pH, to pH 6.50, after which the seawater was left to degas between $CO_2$ injections. Figure 7 details the plot of pH as a function of time for all three sensors. Each individual electrode's pH value has been presented instead of the average of the twelve, in order to improve the tracking of any small pH change. Initially only sensor 2 was tested, and after a day, sensor 1 was incorporated to the same seawater/$CO_2$ tank. Only two of the three sensors were tested in parallel at any one time: sensors 1 and 2 at the beginning, and sensor 3 was swapped with

sensor 1 during the course of the experiment. The data of the three sensors showed favourable agreement between them, confirming the reproducibility between the sensors. They reacted immediately to the pH changes, allowing the end user to

260 detect any spontaneous pH change in the environment. The sensor's measured pHs were validated using a calibrated full ocean pH sensor glass probe by measuring the pH of a small sample at certain times, assuming the efficacy of the glass pH probe after being freshly calibrated. The pH of the sensor and the glass pH probe can only be compared when the seawater was in a stable condition, i.e. before adding CO2 or after letting degas completely. In the first test, before $CO_2$ was injected, sensor 2 presented a pH of 8.39 +/- 0.31 for all individual twelve electrodes over a day in seawater, while the glass pH probe measured

265 a pH of 8.42 when freshly calibrated. The sensors results lay within the specifications outlined with respect to inter sensor repeatability and even though the accuracy cannot be validated against the externally calibrated probe, it can be confirmed than the sensors follow the pH change correctly.

[Figure]

**Figure 7:** A plot of pH as a function of time for three sensors placed in a temperature controlled synthetic seawater solution to periodic injections of $CO_2$, along with samples measured with a calibrated glass pH electrode represented by the open circles.

**4.2 Field Trials**

275 The Field Trials were conducted over a period of three days (8th - 10th September 2021), with a pre-set delay time of 15 min. between measurements. In this case the sensor was placed in autonomous mode, powered directly from a 12 V power supply with no interface control. The data was saved to the internal memory of the sensor and downloaded after deployment. Figure

8 details the data obtained from the sensor through a plot of pH variation (purple) and temperature (black) as a function of time

280 over the 3 days. During the course of the trials, a water sample (ca. 250 mL) was taken in a plastic bottle and a pH of 7.73 was instantly measured using a calibrated glass pH electrode, which was equivalent to the pH 7.74 +/- 0.047 obtained from ANB's pH sensor at that exact time. The pH of the sample was instantly measured in order to have the same temperature conditions as in the sea. The sensor presented a deviation of +/- 0.048 pH units over the entire deployment, which is in accordance with the accuracy stated in the 'technical details' section. The data output from the sensor after 36 h of measuring on a 15 min measurement interval showed the semi-diurnal variations in pH ranging from 7.7 up to 7.9, consistent with the natural ebb and

285 flow of the tidal waterways at Oban. At high tide, the river water mixes with the incoming seawater, producing a higher pH value. The pH variation with tides was quite noticeable, a result of the high tidal coefficient in those 3 days. Tidal coefficients are referred to as the difference in water height between high and low tides, on a scale of 20 to 120. The sensor was deployed from the 8th - 10th September when high tidal coefficients - 101, 99 and 91 occurred (Oban, 2021). Being more specific, the tidal heights in those three days were 0.6 m and 4.3 m for low and high tides on the 8th of September, 0.7 m, 4.1 m, 0.6 m, and

290 4.3 m for the 9th, and 0.8 m and 4.1 m for the last day, 10th of September. All the tidal heights are referenced to the Mean Lower Low Water (MLLW) point, which is the lowest of the two lowest tides per day in the last 19-year period. This phenomenon explains why such significant variations in pH were observed.

[Figure]

**Figure 8:** pH and temperature measurement obtained by the sensor, correlated with high and low tides in Ardmucknish bay, along with the pH of a water sample measured by a calibrated glass electrode (*).

295

During the final day of the trial a glass electrode was deployed alongside the calibration free sensor. The corresponding data comparing the calibration free sensor's response to the glass electrode is shown in Figure 9. It can be clearly seen the glass electrode pH reading was unstable during the course of its deployment, with values ranging between 7.50 and 7.90 in pH, endorsing the advantage of using the calibration-free sensor against the glass electrode for deployment in sea water.

[Figure]

**Figure 9:** A plot of pH as a function of time for the sensors deployed in Ardmucknish Bay for both the calibration free pH sensor and an oceanographic glass-based pH sensor.

**5. Conclusions**

The results demonstrate the use of a new solid-state, calibration-free pH sensor in the monitoring of estuarine and ocean waters. The sensor was shown to respond effectively to pH in laboratory environments whereby the pH of the seawater solution was manipulated by the sporadic injection of $CO_2$ into the mixed solution. Sensors were run in conjunction to each other showing the inter reproducibility of the sensors and compared to a sample of solution whose pH was measured using a freshly calibrated full ocean glass electrode. Favourable agreement was observed throughout the seven-day experiment. The field trials were conducted in an estuarine environment close to Oban in Scotland where the sensor was deployed for a period of three days. The sensor was validated against a sampled solution and tested alongside a glass pH sensor system. The data highlighted the ability of the sensor to monitor the tidal variations of pH in the estuarine environment.

320 **Acknowledgements**

This project has received funding from the European Union's Horizon 2020 research and innovation program under grant agreement No 82297. ANB Sensors gratefully acknowledges funding this work through Innovate UK grant funding, Grant no. 133171. Innovate UK is the UK's innovation agency. It works with people, companies and partner organisations to find and drive the science and technology innovations that will grow the UK-economy. The authors would like to thank the team at

325 SAMS for allowing them to conduct the tests at their facilities.

415 melting. Last access: 10 March 2022.

Why pH is important?: Why pH is important? (aperainst.com), 15 November 2017. Last access: 17 October 2021.

---

## Author Comment (AC5)

**Technical Note: Calibration-Free pH Sensing of Ocean and Estuarine Waters**

Monica Miranda Mugica[1], Christina Day[1], Brandon McHale[1], Kay L. McGuinness[1], Gareth Lee[2], Daisy Pickup[2], Nathan S. Lawrence[1]

[1] ANB Sensors Ltd., 4 Penn Farm, Haslingfield Cambridge, CB23 1JZ, UK
[2] School of Environmental Sciences, University of East Anglia, Norwich Research Park, Norwich, NR4 7TJ, UK

*Correspondence to*: Nathan S. Lawrence (nlawrence@anbsensors.com)

**Abstract.** An electrochemical based, all solid-state, calibration-free pH sensor is presented. The sensor is targeted for monitoring the pH of ocean and estuarine environments covering a salinity range from 10-35  PSU [12] without the need for  any [13] salinity measurements. The sensor performance is demonstrated in both laboratory and field conditions, showing a pH of 8.33 in two sensors, against a measured pH of 8.34 from a freshly calibrated glass pH sensor. Excellent precision is shown, exhibiting a pH of 8.33 +/- 0.015 and 8.33 +/- 0.012 across multiple measurements. The field tests were conducted in an estuarine environment close to Oban in Scotland where the sensor was deployed for a period of three days. The sensor was validated against a sampled solution that measured a pH of 7.74, which was equivalent to the pH 7.74 obtained from ANB's pH sensor at that exact time, and it was tested alongside a glass pH sensor during deployment, demonstrating its stability over 3 days of testing. [11] The data highlighted the  ability [14] 
[revised manuscript text omitted]
 which was calibrated daily using pH 4, pH 7 and pH 10 standard buffers. [31] [37] The results of this are detailed in section 4 'Results and Discussion'.

**3.2 2.2 Field Trials**

The sensor was tested in Ardmucknish bay, located in Dunbeg, a village located outside Oban, Scotland (Figure 6 1). Dunbeg is home to the one of the leading marine science centres in the UK, the Scottish Association for Marine Science (SAMS). [38] Ardmucknish bay is a wide-open, south facing bay, opposite to the island of Mull.

[Figure]

**Figure  1:** Sensor testing location: Ardmucknish bay, Dunbeg, Scotland.

**© Google Maps**

**3 Technology**

To overcome the drawbacks of today's pH sensing devices an innovative approach is presented encompassing a solid-state, calibration-free sensor. The sensor has been developed to be plug and play to the end user, with little maintenance and no need for regular factory recalibrations. It has a range of 2-10 pH units and an accuracy of +/- 0.05 pH units. Figure  2 shows the image of the sensor with the sensing element, or transducer, at the front end. This transducer is a replaceable part which can be easily replaced when required (vide infra), whilst the back end utilises a submersible connector through which power and  communications are passed to the control unit. On board the sensor is the electronic circuitry required to control the sensor and convert the raw measurement signal to the solution pH in a timestamped manner to the end user.

[Figure]

**Figure  2:** (a) The mechanical housing of the sensor including the transducer. (b) An image of the transducer depicting the recessed reference tracking electrodes and the pH sensing electrodes on the front face.

**3.1 Transducer**

This innovative calibration-free pH sensor is based on a voltammetric electrochemical technique, where a time-dependent potential is applied to each sensing electrode in turn and the resulting current is measured as a function of that potential (Bard, 2000). The pH is calculated using the resulting peak potential, as it is linearly associated with the pH [1]. The sensor utilizes a pH active molecule combined with a pH inactive molecule within a solid-state matrix  The electrochemical response of both species [25] can determine the pH of the seawater with no need for calibration, by tracking the performance of the reference electrode (pH inactive molecule) [26] through an additional in-situ electrochemical measurement (Dai et al., 2015a). As mentioned, the peak potential of the pH sensing electrodes moves with pH , allowing the determination of  pH. At the same time, the pH inactive electrode tracks the reference drift,  [27] and therefore, generating a calibration free pH measurement. Figure  3 shows how the peak potential  shifts to lower potentials as the pH of the solution increases for the pH sensing electrodes (Figure 3A) whilst the reference tracker potential stays stable (Figure 3B), only changing in peak position when the reference electrode potential changes. Figure 3 was obtained by testing both pH sensing and reference trackers in synthetic seawater (H2Ocean Natural Reef Salt, Maidenhead Aquatics, UK). The pH of the solution was decreased by the addition of $CO_2$, allowing the response of both the pH sensing and reference sensing elements to be monitored [3]. The pH was measured using a full ocean glass pH probe (Idronaut, Italy), which was calibrated daily using standard buffers [6]. When the pH sensing electrodes and reference tracker are combined, subtracted values can be calculated, and based on the calibration algorithm shown in Fig. 3C, [28] the resulting pH can be determined. The calibration of the sensor shows a sensitivity of 54.8 mV/pH unit in synthetic seawater, overlaying the calibration plot

obtained for pH 4, pH 7 and pH 9 IUPAC standard buffers, with a sensitivity of 56.5 mV/pH unit [1] [10] (Miranda Mugica et al., 2022)

[Figure]

**Figure  3:** The voltammetric response collected by the sensor for both (a) the pH sensing (frequency=100 Hz, step potential=1 mV, amplitude=40 mV) and (b) the reference tracker electrodes (frequency=300 Hz, step potential=0.5 mV, amplitude=80 mV) in synthetic seawater/$CO_2$ system, and (c) the resulting calibration trendline in synthetic seawater and IUPAC standard buffers, using the subtracted values. **[1].** pH values were measured using a calibrated glass pH probe.

The efficacy of this approach is shown in Figure  4. Figure 4A details the variation in peak potential of both the pH sensing and reference tracking systems as a function of time. After 10 h of deployment, a large change in peak potential was observed in the pH sensing system. Nevertheless, the reference tracker exhibited the same change in potential, meaning that the Ag/AgCl redox couple was impacted due to a change in the environment of the reference electrode. [29] The tracking of the reference electrode allows the system to correct the change in the reference electrode potential in-situ.  Figure 4B shows the end user result of pH as a function of time, displaying a pH 8.07 +/- 0.04 over a 14 h experiment, [30]  highlighting the robustness of the sensing system where  no change is seen in the pH measurement owing to the reference electrode tracker correction [3]. Without the reference tracker this phenomenon may be wrongly attributed to a pH change by the end user.

[Figure]

**Figure 3 4**: (a) The plot of peak potential for the reference tracking and pH sensing elements as a function of time. (b) The final calibration free pH sensor output as a function of time.

Figure 4 5 shows the pH values obtained using two sensors tested simultaneously in synthetic seawater in a temperature-controlled tank [6] (see section 2.1), at 288K, where using the recipe outlined provides a pH of 8.30 at 298K. The sensor scans the reference electrode first, to track any changes in the reference electrode potential, and then the twelve pH sensitive electrodes in the transducer head, taking ca.15 seconds for each measurement. [33] This data shows the reproducibility of the two sensors taken in the same tank without any calibration prior to deployment. Across the entire set of measurements an average pH value of 8.33 +/- 0.015 and 8.33 +/- 0.012 for sensor 1 and sensor 2, respectively, were obtained. Without averaging the twelve pH measurements, the accuracy for each sensor was 8.33 +/- 0.04 and 8.33 +/- 0.03. [32] [23] The data obtained is in a perfect excellent [34] agreement with the pH 8.34 measured by a freshly calibrated full ocean glass pH sensor (Idronaut, Italy) as well as with the expected theoretical pH of the synthetic seawater. [31] Even though glass pH electrodes suffer from

reference drift and are not ideal for deployment conditions due to their fragility, a one-point measurement can be trusted when freshly calibrated using standard buffers. [31] [5] For that reason,  the accuracy of the sensor can be confirmed, along with  the reproducibility across different systems.

190

[Figure]

**Figure 4 5**: Representation of pH values obtained by two sensors in synthetic seawater at a constant temperature, compared with the pH measured by a glass pH probe. Each measurement corresponds to the average of twelve  pH measurements over  five minutes testing.

195

**3.2** Lifetime and Maintenance**

The lifetime of the sensor is dependent on the number of measurements that each electrode records and not on the deployment time of the sensor. The electrodes in the transducer are repetitively cycled [35] and  continual voltammetric

200 measurements on each electrode  causes a decay in the response of the electrode. This decay is seen through a loss in the voltammetric peak size until the peak potential can no longer be deciphered. . At this point the sensor notifies the end-user maintenance is required.  It is for this reason that there are a number of electrodes on the front surface of the transducer

205  so as to maximise the lifetime of the sensor before maintenance is required.

The sensor has an onboard processor which  applies the one-time voltammetric sweep parameters to the electrochemical reaction occurring at each electrode. The resulting current received at the electrode due to the electrochemical reaction is then recorded, the data is analysed to determine the potential at which maximum current flows, which is then used to calculate the corresponding pH value. [36] The response of each electrode is analysed after each measurement to ensure an accurate response is observed, when this is not the case the health number is increased, to alert the end user that maintenance is required.

[revised manuscript text omitted]

which the seawater was left to degas between $CO_2$ injections. Figure 7 details the plot of pH as a function of time for all three sensors. Each individual electrode's pH value has been presented instead of the average of the twelve, in order to improve the tracking of any small pH change. Initially only sensor 2 was tested, and after a day, sensor 1 was incorporated to the same seawater/$CO_2$ tank. Only two of the three sensors were tested in parallel at any one time: sensors 1 and 2 at the beginning, and sensor 3 was swapped with sensor 1 during the course of the experiment.[8] The data of the three sensors showed favourable agreement between them, confirming the reproducibility between the sensors. They reacted immediately to the pH changes, allowing the end user to detect any spontaneous pH change in the environment. The sensor's measured pH values were validated using a calibrated full ocean pH sensor glass probe by measuring the pH of a small sample at certain times, assuming the efficacy of the glass pH probe after being freshly calibrated. The pH of the sensor and the glass pH probe can only be compared when the seawater was in a stable condition, i.e. before adding $CO_2$ or after letting it degas completely. In the first test, before $CO_2$ was injected, sensor 2 presented a pH of 8.39 +/- 0.31 for all individual twelve electrodes over a day in seawater, while the glass pH probe measured a pH of 8.42 when freshly calibrated. [3][5]  [40] . The sensors result lay within the specifications outlined with respect to  inter sensor repeatability and even though the accuracy cannot be validaed against the externally calibrated probe, it can be confirmed than the sensors follow the pH change correctly. [41]

[Figure]

**Figure 7:** A plot of pH as a function of time for three sensors placed in a temperature controlled synthetic seawater solution to periodic injections of $CO_2$, along with samples measured with a calibrated glass pH electrode represented by the open circles.

**4.2 Field Trials**

The Field Trials were conducted over a period of three days (8[th] - 10[th] September 2021), with a pre-set delay time of 15 min. between measurements. In this case the sensor was placed in autonomous mode, powered directly from a 12 V power supply with no interface control. The data was saved to the internal memory of the sensor and downloaded after deployment. Figure 8 details the data obtained from the sensor through a plot of pH variation (purple) and temperature (black) as a function of time over the 3 days. During the course of the trials, a water sample (ca. 250 mL) was taken in a plastic bottle and a pH of 7.73 was instantly measured using a calibrated glass pH electrode, which was equivalent to the pH 7.74 +/- 0.047 obtained from ANB's pH sensor at that exact time. The pH of the sample was instantly measured in order to have the same temperature conditions as in the sea.[7] The sensor presented a deviation of +/- 0.048 pH units over the entire deployment, which is in accordance with the accuracy stated in the 'technical details' section. [4] [23] The data output from the sensor after 36 h of measuring on a 15 min measurement interval showed the bi-daily semi-diurnal variations in pH ranging from 7.7 up to 7.9, consistent with the natural ebb and flow of the tidal waterways at Oban. At high tide, the river water mixes with the incoming seawater, producing a higher pH value. The pH variation with tides was quite noticeable, a result of the high tidal coefficient in those 3 days. Tidal coefficients are referred to as the difference in water height between high and low tides, on a scale of 20 to 120. The sensor was deployed from the 8[th] - 10[th] September when high tidal coefficients - 101, 99 and 91 occurred (Oban, 2021). Being more specific, the tidal heights in those three days were 0.6 m and 4.3 m for low and high tides on the 8[th] of September, 0.7 m, 4.1 m, 0.6 m, and 4.3 m for the 9[th], and 0.8 m and 4.1 m for the last day, 10[th] of September. All the tidal heights are referenced to the Mean Lower Low Water (MLLW) point, which is the lowest of the two lowest tides per day in the last 19-year period. [42] This phenomenon explainings why such significant variations in pH were observed.

[Figure]

310

**Figure 8:** pH and temperature measurement obtained by the sensor, correlated with high and low tides in Ardmucknish bay, along with the pH of a water sample measured by a calibrated glass electrode (*).

315  During the final day of the trial a glass electrode was deployed alongside the  calibration free sensor. The corresponding data comparing the  calibration free sensor's response to the glass electrode is shown in Figure 9. It can be clearly seen the glass electrode  pH reading was unstable during the course of its deployment, with values ranging between 7.50 and 7.90 in pH, endorsing the advantage of using  the calibration-free sensor against the glass electrode for deployment in sea water.

[revised manuscript text omitted]

Martz, T.R., Daly, K.L., Byrne, R.H., Stillman, J.H. and Turk, D.: Technology for Ocean Acidification Research, Oceanogr., 28, 40-47, doi:10.5670/oceanog.2015.30, 2015. [20]

Martz, T.R., Gonnery, J.G. and Johnson, S.: Testing the Honeywell Durafet® for seawater pH applications, Limnol Oceanogr.: Methods, 8, 172-184, doi:10.4319/lom.2010.8.172, 2010.

400 Miranda Mugica M., McGuinness, K.L. and Lawrence, N.L.: Electropolymerised pH insensitive Salicylic acid reference systems: utilization in a novel pH sensor for food and environmental monitoring, Sensors, 22, 1-12, doi:10.3390/s22020555.

Newhall, K., Krishfield, R., Peters, D. and Kemp, J.: Deployment Operation Procedures for the WHOI Ice-Tethered Profiler, Technical report, Woods Hole Oceanographic Institution Woods Hole, MA, 41 pp., 2007.

Oban: Tydes and solunar charts: Tide times and charts for Oban, Scotland and weather forecast for fishing in Oban in 2021 (tides4fishing.com). Last access: 30 October 2021.

Ocean acidification: Ocean acidification | National Oceanic and Atmospheric Administration (noaa.gov), 1 April 2020. Last access: 17 October 2021.

pH – SeaBed: PH – Seabed. Last access: 20 October 2021.

Rerolle, V.M.C., Floquet, C.F.A., Harris, A.J.K., Mowlem, M.C., Bellerby, R.R.G.J. and Achterberg, E.P.: Development of a colorimetric microfluidic pH sensor for autonomous seawater measurements, Analytica Chimica Acta, 786, 124-131, doi:10.1016/j.aca.2013.05.008, 2013.

Rerolle, V.M.C., Ruiz-Pino, D., Rafinadeh, M., Loucaides, S., Papadimitrious, S., Mowlem, M. and Chen, J.: Measuring pH in the Arctic Ocean: Colorimetric method or SeaFET?, Methods in Oceanogr., 17, 32-49, doi:10.1016/j.mio.2016.05.006, 2016.

Sabine, C.L., Feely, R.A., Grumber, N., Key, R.M., Lee, K., Bullister, J.L., Wanninkhof, R., Wong, C. S., Wallace, D. W. R., Tilbrook, B., Millero, F.J., Peng, T.H., Kozyr, A., Ono, T. and Rios, A.F.: The oceanic sink for anthropogenic CO2, Science, 305:5682, 367-71, doi:10.1126/science.1097403, 2004.

Seidel, M.P., DeGrandpre, M.D. and Disckson, A.G.: A sensor for in situ indicator-based measurement of seawater pH, Marine Chemistry, 109, 18-28, doi: 10.1016/j.marchem.2007.11.013, 2008. [22]

Shitashima, K., Kyo, M., Koike, Y. and Henmi, H.: Development of in situ pH sensor using ISFET, IEEE 2002 International Symposium on Underwater Technology - Tokyo, Japan, doi:10.1109/UT.2002.1002403, 2002. [21]

Siegenthaler, U., Monnin, E., Kawamura, K., Spahni, R., Schwander, J., Stauffer, B., Stocker, T.F., Barnola, J.M. and Fischer, H.: Supporting evidence from the EPICA Dronning Maud Land ice core for atmospheric $CO_2$ changes during the past millennium, 57:1, 51-57, doi:10.3402/tellusb.v57i1.16774, 2017.

Sonune, A. and Ghate, R.: Developments in wastewater treatment methods, Desalin., 167, 55-63, doi: 10.1016/3.desal.2004.06.113, 2004. [15]

Takeshita, Y., Martz, T.R., Johnson, K.S. and Dickson, A.G.: Characterization of an Ion Sensitive Field Effect Transistor and Tilton, F., Benson, W.H. and Schlenk, D.: Evaluation of estrogenic activity from a municipal wastewater treatment plant with predominantly domestic input, Aquat. Toxicol., 61, 211-224, doi:10.1016/s0166-445x(02)00058-9, 2002. [15]

Underwater measurement solutions: Sea & Sun Technology - Multiparameter Probes (sea-sun-tech.com). Last access: 20 October 2021.

Why pH is important?: Why pH is important? (aperainst.com), 15 November 2017. Last access: 17 October 2021.